# Nanoparticle-Aided Detection of Colorectal Cancer-Associated Glycoconjugates of Extracellular Vesicles in Human Serum

**DOI:** 10.3390/ijms221910329

**Published:** 2021-09-25

**Authors:** Rufus Vinod, Randa Mahran, Erica Routila, Janne Leivo, Kim Pettersson, Kamlesh Gidwani

**Affiliations:** 1Department of Life Technologies, University of Turku, 20520 Turku, Finland; rufvin@utu.fi (R.V.); randa.r.mahran@utu.fi (R.M.); eamkuu@utu.fi (E.R.); jpleiv@utu.fi (J.L.); kim.pettersson@utu.fi (K.P.); 2Tropical Health and Parasitology Department, High Institute of Public Health, Alexandria University, Alexandria 21617, Egypt

**Keywords:** colorectal cancer, extracellular vesicles, CD151, CD63, europium nanoparticles, glycovariant assay

## Abstract

Extracellular vesicles (EVs) are found in all biological fluids, providing potential for the identification of disease biomarkers such as colorectal cancer (CRC). EVs are heavily glycosylated with specific glycoconjugates such as tetraspanins, integrins, and mucins, reflecting the characteristics of the original cell offering valuable targets for detection of CRC. We report here on europium-nanoparticle (EuNP)-based assay to detect and characterize different surface glycoconjugates of EVs without extensive purification steps from five different CRC and the HEK 293 cell lines. The promising EVs candidates from cell culture were clinically evaluated on small panel of serum samples including early-stage (*n* = 11) and late-stage (*n* = 11) CRC patients, benign condition (*n* = 11), and healthy control (*n* = 10). The majority of CRC cell lines expressed tetraspanin sub-population and glycovariants of integrins and conventional tumor markers. The subpopulation of CD151 having CD63 expression (CD151^CD63^) was significantly (*p* = 0.001) elevated in early-stage CRC (8 out of 11) without detecting any benign and late-stage samples, while conventional CEA detected mostly late-stage CRC (*p* = 0.045) and with only four early-stage cases. The other glycovariant assays such as CEA^Con-A^, CA125^WGA^, CA 19.9^Ma696^, and CA 19.9^Con-A^ further provided some complementation to the CD151^CD63^ assay. These results indicate the potential application of CD151^CD63^ assay for early detection of CRC patients in human serum.

## 1. Introduction

Diagnosing colorectal cancer (CRC) at an early stage is the key to reduce mortality and treating early-stage lesions. The 5-year survival rate is 90% for 39% of patients diagnosed with a localized tumor, whereas this percentage declines to 71% and 14% with regional and distant stages, respectively [1]. The available screening tests are based on either detection of occult blood or tumor DNA in stool or on visualization of the tumor by colonoscopy, sigmoidoscopy, or colonography. Among all tests, colonoscopy and sigmoidoscopy have the highest sensitivity and specificity; however, it is considered an invasive procedure and involves an unpleasant preparation process [2].

Carcinoembryonic antigen (CEA) is the most widely accepted blood-based biomarker for CRC, which has proved valuable for monitoring [3]. However, high concentrations are also found in benign conditions such as hepatitis, pancreatitis, inflammatory bowel disease, and Crohn’s disease [4,5] as well as in a wide range of other malignant carcinomas, e.g., pancreatic, lung, ovarian, and gastric cancer. The inadequate specificity of CEA impedes its use for early-stage CRC diagnosis and disease progression. For this reason, supplementary biomarkers to CEA, such as cancer antigen 19.9 (CA19.9) [6], cancer antigen 125 (CA125) [7], cancer antigen 72-4 (CA72-4) [8], and tumor-associated glycoprotein 72 (TAG 72) [9], have also been recognized and suggested as aids in CRC diagnosis, post-operative surveillance, as well as monitoring response to treatment [3,5]. However, there is still no serological biomarker that proves to provide sensitive, specific, and meaningful information for CRC diagnosis and treatment outcomes in a comprehensive way.

Extracellular vesicles (EVs) are nanoscale-sized particles released by all body cells under normal physiological conditions and have the same topology as the plasma membrane [10,11]. EVs play a major role in reflecting cell communication and thus help unravel the pathogenesis and progression of many diseases. Nowadays, there is an increasing appreciation for EVs as possible diagnostic and therapeutic targets. It is, for example, claimed that EVs are released in oxidative stress and hypoxia; therefore, it can be proposed for the treatment of pulmonary artery hypertension (PAH) and can be used as a diagnostic marker for measuring oxidative stress [12,13]. Another promising and recent application of EVs is in cardiovascular diseases (CVD) through functional characterization of EVs secreted in cells of the cardiovascular system [14]. They also hold great potential for the identification of disease biomarkers, for example, in cancer since cells release EVs that are found in body fluids, such as plasma, urine, or cerebrospinal fluid; therefore, they are considered surrogates of the originating cells and the amount of cancer-derived EVs is usually double than the normal [15,16,17,18,19]. Cancer-cells-derived EVs also carry cargo, enriched with integral member glycoconjugates such as tetraspanins (CD9, CD63, CD81, and CD151), integrins α and β, EpCAM, and mucins [20,21,22] having more aberrant glycosylations than the normal physiological conditions [21]. Recently, EVs from CRC cells, which were CEA positive, were found to induce morphological and functional changes in colonic mesenchymal stromal cells, possibly favoring tumor growth and malignant progression [23]. Exosomal CD 151 acts as a principal promoter of metastasis, and its expression and glycosylation have been identified in the CRC-derived EVs [24]. The current standard methods to quantify specific protein expression on EVs are immune electron microscopy and flow cytometry using fluorescence threshold triggering, yet their routine application is prohibited due to the tedious procedures and limited statistical power (Ye Tian 2018).

Changes in protein glycosylation are universally seen in different types of cancers; therefore, the study of glycoconjugates of EVs has gained massive interest, as it opens the door for the identification of a novel range of glycovariant based cancer biomarkers with sensitivity and specificity that overcome the existing biomarkers [25,26]. Changes in N- and O-glycosylation is well studied and have been reported in many cancer EVs, such as altered glycosylation of O-glycans is a characteristic change in the mucins, which dissipates the normal maturation of the glycans and leads to the overexpression of Thomsen-Friedenreich-related T, Tn, and STn antigens [27], while aberrant glycosylation of N-glycans, proximal fucosylation, and the corresponding enzyme FUT8 are upregulated in several types of cancer.

The identification and quantification of EVs in clinical samples remains challenging. Previously, we detected and characterized glycoconjugates of EVs by using an ultra-sensitive reporter, fluorescent europium-nanoparticles (Eu NPs)-based sandwich assays using a lectin- or glycan-binding antibody or EVs associated antibody from cell culture supernatant, serum, or urine samples without any extra steps of purification [28,29,30].

In the present study, we utilize the Eu NPs approach to develop simple and robust EV-based assays with utility for CRC. Protein epitope detecting anti-mucins, -integrin, and -tetraspanin antibodies were immobilized on microtiter, and panel of lectin- or EVs- related antibodies coated onto Eu NPs were tested for the detection of immobilized antigens from cell-culture-spent medium of five different CRC cell lines, and HEK 293 cell line was used as a negative control. The discovered promising candidates were then validated with serum samples of CRC, including early and late stages; the cancer samples were tested side by side with benign disease and healthy controls and compared with conventional CEA, CA19-9, and CA125 immunoassays.

## 2. Results

### 2.1. Screening of Novel Glycoconjugates

With the use of ten different protein epitope detecting antibodies of EVs-related proteins (mucins, integrins, and tetraspanins) used as biotinylated capture antibodies and a panel of lectins and anti-EVs antibodies coated onto Eu NP as tracers, we observed the different binding patterns to the five different CRC, and one control cell culture spent medium (CCSM), as shown in Figure 1. The combinations of the biotinylated capture antibodies and the NP tracers were denominated as antigen^lectin/mAB^, e.g., CEA^Ma695^ and/or CEA^CD63^.

Several combinations have been found, which seems to be promising, and the signal-to-background (S/B) ratio of 9.0 has been considered to be the significant binding of glycan specific conjugates to the immobilized antigen. All the cancerous CCSM showed significant binding with many combinations except Colo 320 DM, which is a unique cell line representing the less common neuroendocrine type of CRC. The biotinylated anti-CD 63 and anti-CD 81 antibodies coated onto NPs (CD63^CD81^) and vice versa, i.e., CD81^CD63^ in Colo 320 DM showed very high expression with S/B of 517.0 and 73.4, respectively. Macrophage galactose-type lectin (MGL) was the only lectin that showed binding to CD 63 and CA125 with S/B of 18.6 and 16.4, respectively. Similarly, CD151 bound to CD147 in Colo 320 with S/B of 16.5.

The expression of CA19.9 (C192 antibody) was detected with anti-CD151, -ITGA-6, -ITGB-4, and -CD147 capture antibodies on most of the CRC CCSM compared to the control. Among the tested five CRC cell lines, many combinations clearly had a strong binding affinity with early-stage cell line LS17 4T. With an immobilized CD151, the binding with anti-CD 63 NP was significantly elevated in four CRC cell lines with no significant binding in the control cell line. The best 37 combinations were selected for clinical evaluation on the serum cohort (*n* = 42).

### 2.2. Comparison of Conventional EIA with Eu NPs TRF Assay

Concentrations of CEA, CA19.9, and CA125 were analyzed in 42 serum samples by the EIA analysis (Fujirebio Diagnostics, Gothenburg, Sweden) according to the manufacturer’s instructions. CA19.9 EIA and CA125 EIA recognize most of the late-stage samples, with less discrimination in healthy + benign vs. early-stage samples. The discriminative ability of CEA in healthy + benign vs. early-stage CRC samples was (*p =* 0.04572), whereas for healthy + benign vs. early and late-stage CRC samples it was also significant (*p =* 0.05052), as shown in Figure 2. Among all the combinations tested in serum samples based on data from CRC cell line glycoprofiling, CD151^CD63^ was the most promising combination; it outperformed the conventional CEA EIA and showed a significant difference between healthy + benign vs. early-stage CRC samples (*p* < 0.00183). This demonstrates that there is a high expression of transmembrane glycoprotein CD151 on the exosomes secreting in early-stage CRC patients, whereas the late-stage samples do not show any expression of CD151. 

### 2.3. Glycoconjugates of EVs Analysis in Clinical Serum Samples

The sensitivity and specificity of the conventional and EVs-based assays and their complementation to the CD151^CD63^ assay are shown as a heat map in Figure 3. The concentration of the conventional biomarkers (CA125, CA 19.9, and CEA) and signal-to-background ratio of the EVs assay were used for classifying the healthy, benign, early-, and late-stage CRC samples. The glycovariant assays such as CEA^Con-A^, CA125^WGA^, CA 19.9^Ma696^, and CA 19.9^Con-A^ recognized most of the late-stage samples, while CD151^CD63^ assay detected only early-stage CRC sample (8 out of 11). The two early-stage negative samples with CD151^CD63^ were positive with the CEA EIA, CEA^Con-A^, CA125^WGA^, and CA19.9^Con-A^.

## 3. Discussion

Extracellular vesicles (EVs) have been widely investigated in recent years, and emerging data suggest that cancer-derived EVs are strongly glycosylated, being rich in specific glycoconjugate [21]. In this study, we report on fluorescence europium chelate dyed-nanoparticles (Eu-NP)-based TRF assays to demonstrate the presence of specific proteins and glycans on the surface of EVs in CRC cell culture spend medium and further clinical evaluation of promising candidates in clinical serum samples–We have previously utilized the lectin-NP-based platform successfully to explore the glycosylation of serum glycoproteins CA125 (CA125^MGL^) and CA15-3 (CA15-3 WGA) in ovarian and breast cancer patients’ samples [31,32]. More recently, our group developed NP-based TRF assay for analysis and characterization of EVs and identification of disease-specific markers on the surface of patient-derived urinary EVs [30,33]. Our NP-TRF assays have the advantages of being: (1) robust, simple, and sensitive for detection and characterization of EVs from circulation without any purification steps; (2) cancer-associated EVs can be detected through the protein or glycan moieties presented on their surface glycoconjugates using lectins or glycan-binding antibodies as binders; (3) signal amplification provided by thousands of Eu chelates doped in a single particle; and (4) the strengthening of the functional affinity (avidity) of the lectins to their target glyco-structure epitopes enabled by the high-density immobilization of lectin on the particles.

Several EV-related assay candidates were elevated in CRC associated cell culture compared to control. The subpopulation of tetraspanin CD151, also having CD63 expression (CD151^CD63^), was detected in the majority of early-stage CRC serum samples (8 out of 11), without being detected in any of the late-stage CRC, benign, and healthy controls. Although the conventional CEA EIA significantly discriminate (*p* < 0.05) CRC samples from benign and healthy control, it is elevated only in 4 out of 11 early-stage CRC with only one early-stage serum showing above conventional cut-off (5 ng/mL) of CEA EIA. As for late-stage, 5 out of 11 were positive, with all having more than 5 ng/mL. The CD151^CD63^-negative samples were further complemented with the glycovariant assays CEA^Con-A^, CA19.9^Con-A^, and CA125^WGA^ along with CEA EIA to detect all the early-stage (10 out of 11) and late-stage (8 out of 11) CRC serum samples.

Using tetraspanin CD151 antibody-based immunohistochemistry, Lin et al. *(16)* found dynamic changes in the expression of this tetraspanin, observed in 48% of patients with early-stage CRC versus only 33% of patients with metastatic colon cancer [34]. Similarly, Hashida et al. (2003) reported positive CD151 membrane stain in approximately 55% of patients with early-stage CRC. The 3-year survival rate of patients with CD151-positive tumors was significantly lower than that of patients with CD151-negative tumors [35]. In our study, EVs having CD151 with CD63 detect the majority of early stage without detecting any late stage in a simple, robust immunoassay in circulation. Furthermore, CD151 also forms a complex with the integrins (*α*3 *β*1, *α*6 *β*1, and *α*6 *β*4) to drive the metastatic signaling pathways in the cell to promote neovascularization [36]. CD63 has also been implicated in the transport and regulation of other proteins in the progression of melanoma. Rank et al. reported the elevation of CD63 expression in plasma of ovarian cancer [37]. Kaprio et al. reported in their study that CD63 execute immunohistochemistry on epithelial–mesenchymal transition in CRC patients having higher expression associated with adverse effects [38]. Furthermore, Miki et al. and Lewitowicsz et al. also reported the positive correlation of CD63 higher expression with the poor survival rate of gastric and gastrointestinal stromal tumor [39,40]. In relation to these studies, we have identified the CD63 expression on CD151 expression in serum of early-stage CRC patients.

ITGA2, ITGA6, and ITGB4 through their EV complexes from immobilized CRC cell lines bound most effectively to the mannose-binding lectin (MBL) and fucose-binding lectin, Aleuria Aurantia (AAL). The bound lectins highlighted the previously suggestions concerning the *N*-glycosylation, which is the most remarkable change in cancer, which involves the increased branching of *N*-glycans and the proximal fucosylation [21,41]. Beaulieu et al. reported the upregulation of ITG A6 and B4 subunits in CRC signaling to promote cell migration, invasiveness, and underlying malignancies [42]. The ITGA2 has also previously been proposed to be a biomarker of progressing CRC [43]. However, in our work, the glycovariants of ITGA2, ITGA6, and ITGB8 did not work in the clinical serum analysis. In a previous study, we reported fucosylated glycoisoforms of ITGA3 directly from unprocessed urine to distinguish bladder cancer from age-matched benign controls in a simple sandwich assay [33].

Several lectins bound to the CRC-associated conventional biomarkers such as CEA, CA125, and CA19.9 in the cell line screening. The concanavalin A (Con-A) lectin binds to the mannose sugar in the *N*-glycosylation, and its binding patterns change in the adenoma–carcinoma sequence [44]. In the current study, Con-A demonstrated binding to C19.9 and CEA CRC samples; CA19.9^Con-A^ detected 11 out of 22 CRC samples, whereas CEA^Con-A^ detected 6 out of 22 CRC samples, but interestingly the two negative early-stage samples with CD151^CD63^ were found positive with these glycovariant assays. WGA significantly binds to the *N*-acetylglucosamine and sialic acid associated with the carbohydrate structures, which are altered during the malignant progression [45]. In our study, CA125^WGA^ and CA19.9^Ma695^ also recognized 11 and 9 out of 22 CRC samples, respectively, and complemented the CD151^CD63^ assay. Two possible explanations for the discordance between the cell culture and serum in the EVs assays candidates is that the particular biomarker candidate does not exist in vivo or present in minute amount and with matrices interference, representing technical artifacts.

This study was designed as a pilot evaluation of a prototype diagnostic test where we have used heterogenous samples to cover all possible stages, histology (adenocarcinoma and mucinous), and locations (ascending, descending, transverse colon, rectum, and sigmoid) of CRC and used a limited number of purposefully selected individuals [46]. The major limitation of this study was the cohort size; hence, further validation is required on a larger number of subjects. Moreover, CD151^CD63^ assay detects only early-stage CRC; this novel assay in combination with other glycovariant (CEA^Con-A^, CA125^WGA^, and CA 19.9^Con-A^) could be used for follow-up of CRC patients post-treatment for detection of early relapse as well as screening of CRC.

## 4. Materials and Methods

### 4.1. Clinical Samples

Serum samples (*n* = 10) from healthy volunteers were included from Turku University, Department of Biotechnology (Turku, Finland), and serum samples (*n* = 10) from Benign patients having Crohn’s Disease and serum samples (*n* = 22) from CRC patients (*n* = 11) early stage I–II and (*n* = 11) late stage III–IV were purchased from Discovery Life Sciences, Inc. (Huntsville, AL, USA). The age of healthy volunteers ranged from 24 to 42 years, the age of the benign patients from 44 to 69 years, and the age of CRC patients from 46 to 84 years. The median ages were 37, 59.5, and 70 years for the healthy, benign, and CRC samples, respectively. Patients’ demographic characteristics are presented in Table 1, whereas the clinicopathological characteristics of CRC patients are presented in Table 2. Most of the cancer samples were taken before the start of cancer treatment, all were of non-Hispanic origin, and none of them had a history of benign disease.

### 4.2. Reagents and Equipment

The 96-well microtiter plates coated with streptavidin (SA plates, product #: 41-07TY), wash buffer (product #: 42-01TY), and RED assay buffer (product #: 42-02TY) were purchased from Kaivogen Oy (Turku, Finland). From R&D systems, we acquired the anti-integrin’s antibody and anti-tetraspanins’ antibody, e.g., anti-CD151 (Mab 1884). The anti-CD63 (H5C6 clone) antibody was purchased from BD Science (Vantaa, Finland). The plate washer (Delfia PlateWash 1296-026) and plate shaker (Delfia PlateWash 1296-026) were from Wallac Oy (Turku, Finland), and HIDEX™ fluorimeter was from HIDEX Oy (Turku, Finland). The reagents for cell line cultures were Gibco brand of Thermo Fisher Scientific (Waltham, MA, USA) except for glutamine, which was Ultraglutamine from Lonza (Basel, Switzerland), and phosphate-buffered saline, which was from GE Healthcare (Chicago, IL, USA). The conventional CEA, CA125, and CA19-9 EIA kits were kindly provided by Fujirebio Diagnostics.

### 4.3. Cell Cultures

The human CRC cell lines Colo 205, Colo 320 DM, SW 403, LS17 4T, and SW 1463 were cultured in Dulbecco’s Modified Eagle Medium (DMEM), supplemented with 10% inactivated fetal bovine serum (FBS), 1% glutamine, and 1% penicillin-streptomycin. The cell culture spent medium (CCSM) was kindly gifted by Fujirebio Diagnostics. The human embryonic kidney 293 cell line HEK293 was cultured in Expi293 medium, commercially purchased from Thermo Fischer Scientifics (Waltham, MA, USA). The cells were cultured at 37 °C under 5% CO_2_. When the cells reached the confluence of approximately 70%, the medium was collected and centrifuged for 3 min at 161× *g*. The CCSM was collected and stored at −80 °C. The spent medium was then concentrated 5 times with Vivaspin Turbo 15 filter (Sartorius Stedim Lab Ltd., Stonehouse, UK) and stored at −20 °C.

### 4.4. Preparation of Nanoparticle–Bioconjugates

The Eu-NPs as bioconjugated reporter molecules have been described previously [47]. Amino groups of glycan-binding proteins (lectins), tetraspanins, and integrin monoclonal antibodies (mABs) were covalently bound to the activated carboxyl group of the Eu NPs according to the previously described procedure [28,32]. Briefly, N-hydroxysulfosuccinimide and N-(3-dimethylaminopropyl)-N′-ethylcarbodimide were incubated with 1e10^12^ NPs in 50 mM 2-(N-morpholino) ethanesulfonic acid (MES) buffer (pH 6.1) at final concentrations of 10 and 0.75 mmol/L, respectively, at room temperature for 15 min. The concentration of lectins and mAbs in the coupling reactions was 0.625 g/L. The coupling reactions were incubated for 2 h at room temperature under continuous mixing. Final washes and blocking of the remaining active groups were performed in Tris buffer (10 mmol/L Tris, 0.5 g/L NaN_3_, pH 8.5), and the NPs-protein conjugates were stored in the same buffer supplemented with 2 g/L BSA (Bioreba) at 4 °C. Before every use, particles were vortex mixed thoroughly and sonicated to disperse any large aggregates.

### 4.5. Biotinylation of Antibodies and Preparation of Solid-Phase Surfaces

The solid-phase different anti-tetraspanin, anti-integrins, and anti-mucins antibodies were biotinylated for 4 h at room temperature with a procedure described earlier (Gidwani et al. 2016). Biotinylated antibodies were purified with NAPTM-5 and NAPTM-10 gel-filtration columns by use of 50 mmol/L Tris-HCl (pH 7.75), containing 150 mmol/L NaCl and 0.5 g/L NaN_3_. The labeled antibody was stored in 1 g/L BSA at 4 °C.

### 4.6. In-House Time-Resolved Fluorometry Assay for Cancer-Associated Glycan

In-house time-resolved fluorometry (TRF) immunoassays for antibodies against different EVs related glycoconjugates (tetraspanins, integrins, and mucins) with the panel of lectins/mABs coated onto Eu NPs were used for the glycoprofiling of cancer-associated glycans in CRC cell lines have already been described [28,30]. All the incubations were performed at room temperature, and wash steps were done by using the Kaivogen wash buffer with the dilution of 1× concentration. Biotinylated capture tetraspanins, integrins, and mucins (monoclonal antibodies) with the concentration of 50 ng/25 μL were immobilized to streptavidin-coated low-fluorescence microtiter wells in the assay buffer for 60 min without shaking. After washing the wells twice, 20 μL of RED assay buffer was added followed by the 5 μL of DMEM and CCSM from each cell line or diluted serum samples. The plates were incubated for one hour in slow shaking and washed two times. After washing, 1 × 10^7^ lectin/mAB coated onto Eu–NPs were added per well in 25 μL of RED assay buffer supplemented with CaCl_2_ for MGL (macrophage galactose-type lectin), DC-SIGN (dendritic cell-specific intercellular adhesion molecule-3–grabbing non-integrin) and MBL (mannan-binding lectin) and incubated for 90 min with shaking. The wells were washed six times, and the TRF of Eu^+3^ was measured with HIDEX fluorometer (λ_ex_ = 340 nm and λ_em_ = 615 nm), as shown in Figure 4.

### 4.7. Screening of Glycoconjugate Biomarkers in Human Serum Samples

The screened TRF assays from glycoprofiling of CCSM were performed on the human serum samples to screen the novel biomarkers to discriminate the healthy, benign, and CRC. The exception was that serum samples were diluted in 1:10 concentration in Red assay buffer. TSA-BSA was used as a blank. The conventional CEA, CA125, and CA19.9 were measured using the Fujirebio Diagnostic enzyme immunoassay kit according to the manufacturer’s instructions.

### 4.8. Statistical Analyses

The results from the glycovariant biomarker screening from the CCSM and serum samples were plotted using RStudio [48] software with ggplot2 [49] R packages. Marker concentrations in disease groups were compared using Kruskal–Wallis one-way ANOVA with post hoc Dunn test on rank-transformed data. Statistical analyses were performed using SigmaStat software (Systat Software). Statistical difference was considered significant if *p*-value was 0.05. For biomarker values, normality was evaluated applying Shapiro–Wilk test, and it was also assessed visually. Medians and 25th/75th quartile for conventionally measured CEA, CA125, CA19.9, and CD151^CD63^ were calculated in different diagnostic groups. Leven’s test was applied to assess the equality of the variances. Heatmap analysis using ggplot 2 [49] was performed to further evaluate the distribution of the samples subgroups in the CEA EIA, CA125 EIA, CA19.9 EIA, CEA^Con-A^, CA125^WGA^, CA19.9^Ma695^, CA19.9^Con-A^, and CD151^Cd63^.

## 5. Conclusions

This study suggests that Eu NPs-based CD151^CD63^ assays in combination with glycovariant of CEA and CA125 are more sensitive in distinguishing CRC patient’s serum from benign and healthy controls than conventional CEA biomarkers alone. After further validation study, the combination of CD151^CD63^, CEA^Con-A^, CA19.9^Con-A^, and CA125^WGA^ assays could be used in differential diagnosis and early detection of CRC disease.

## Figures and Tables

**Figure 1 ijms-22-10329-f001:**
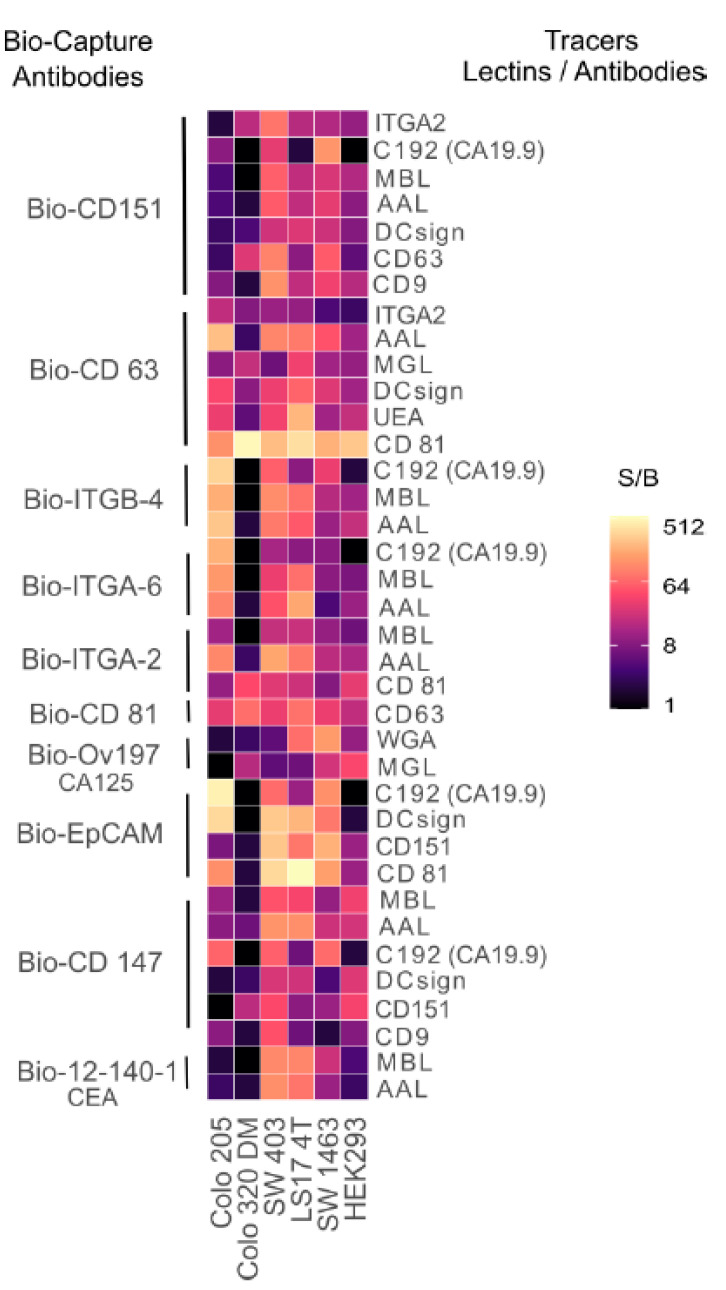
Detection and characterization of glycoconjugates of extracellular vesicles. Signal-to-background (S/B) ratio of the glycoconjugates of EVs-based assays from cell culture spent medium of five different CRC and human embryonic kidney 293 as negative control.

**Figure 2 ijms-22-10329-f002:**
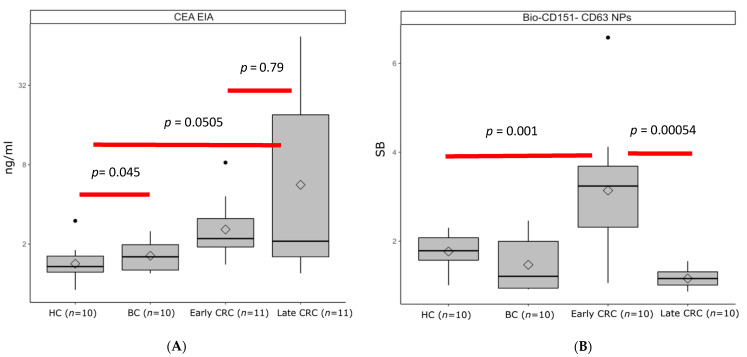
Discrimination of CRC early- (*n* = 11), late-stage (*n* = 11) from benign (*n* = 11) and healthy controls (*n* = 10) using (**A**) conventional CEA EIA and (**B**) CD151CD63 assay. CEA in early stage (*p* = 0.04) and late stage (*p* = 0.05) were significantly higher than in healthy and benign controls and significant different (*p* = 0.79) between early and late stage, while the CD151CD63 was significantly higher only in early-stage (*p* = 0.001) CRC samples.

**Figure 3 ijms-22-10329-f003:**
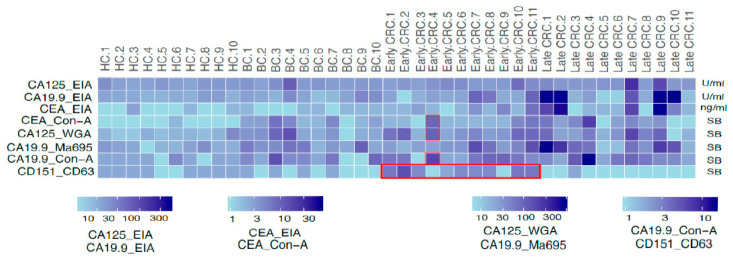
Conventional EIA levels and glycoconjugates of EVs. The concentration (U/mL) of the conventional biomarkers (CA125, CA19.9, and CEA EIA) and their glycovariants (CA19.9Con-A, CEACon-A, and CA125WGA) and CD151CD63 assay. Among conventional EIA, CEA performed best and was mostly elevated in late-stage CRC, while CD151CD63 assay was specifically elevated in early stage; it detected highest number of early-stage CRC patients (9 out of 11) and was further complemented by glycovariant assays.

**Figure 4 ijms-22-10329-f004:**
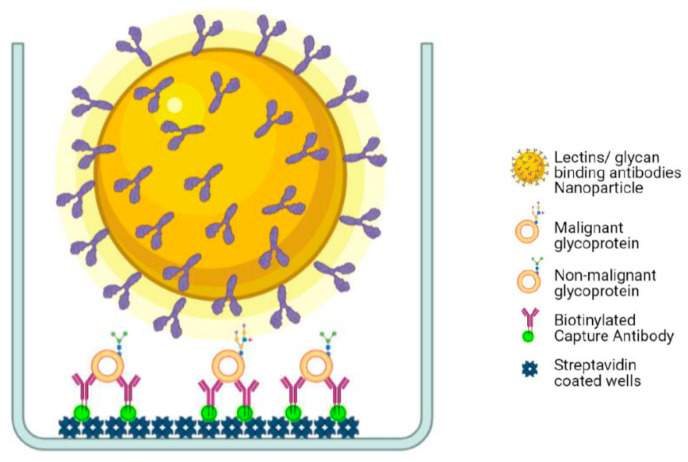
The illustration of the principle of Eu NP-based time-resolved fluorometry assay for detection of glycoconjugates of EVs. A sandwich assay, in which protein epitope recognizing mab-captured different glycoconjugates of EVs from the cell-culture supernatant or clinical serum samples is traced with either protein (by MAb) or glycan moieties by lectins coated on the surface of Eu-NPs (www.biorender.com, accessed on 24 September 2021).

**Table 1 ijms-22-10329-t001:** Patients’ demographic characteristics.

Demographic Data	Age at Collection (Years) Mean (Min–Max)	Gender		Ethnicity
		Male	Female	Non-Hispanic
Healthy volunteers (*n* = 10)	37 (24–42)	7 (70%)	3 (30%)	10 (100%)
Benign control (Crohn’s) (*n* = 10)	59.5 (44–69)	6 (60%)	4 (40%)	10 (100%)
CRC patients (*n* = 22)	68 (44–84)	15 (68%)	7 (32%)	22 (100%)

**Table 2 ijms-22-10329-t002:** Patients’ clinicopathological characteristics.

CLINICOPATHOLOGICAL CHARACTERISTICS OF CRC	CRC Patients (*n* = 22)
**HISTOPATHOLOGICAL DIAGNOSIS**	
Adenocarcinoma	11 (50%)
Mucinous adenocarcinoma	2 (9%)
Undefined	9 (41%)
**LOCATION**	
Proximal *	3
Distal **	4
Rectum	2
Unknown	12
**CLINICAL STAGE I-IV**	
*I*	5 (22.7%)
*II*	6 (27.3%)
IIA	3
II-B	2
II-C	1
*III*	6 (27.3%)
III-A	3
III-B	3
IV	5 (22.7%)
**TNM STAGING**	
*T*	
Tx	1
T1	-
T2	1
T3	4
T4	1
*N*	
Nx	2
N0	2
N1	3
N2	-
*M*	
M0	16
M1	5
**TREATMENT STATUS**	
Pre-treatment	8 (36.4%)
Post treatment	6 (27.2%)
Active treatment	4 (18.2%)
Unknown	4 (18.2%)

Proximal *—caecum and transverse column); distal **—descending column and sigmoid.

## Data Availability

All datasets generated during the current study are available from the corresponding author on a reasonable request.

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
