# Peer review of "Nanoparticle-Aided Detection of Colorectal Cancer-Associated Glycoconjugates of Extracellular Vesicles in Human Serum"

_ijms, 2021, doi:10.3390/ijms221910329_

Round 1

Reviewer 1 Report

Rufus Vinod et al describe a europium-nanoparticle (EuNP)-based assay to detect and characterize different surface glycoconjugates of EVs without extensive purification steps from five different CRC and the HEK 293 cell-lines. Overall this is an interesting attempt for a potential application for early detection of CRC patients in human serum. But, I noticed some typing errors in the bibliography (please merge the numbers) and check the spaces of the words.

Introduction:

Discuss a little bit more the potential of extracellular Vesicles to be used for different applications such as to measure oxidative stress (recommended reference: https://pubmed.ncbi.nlm.nih.gov/32354089/

Results:

Figure 2: correct 0,79 with 0.79

How do you explain the high standard deviation noticed for late CRC?

Do you believe a limit of the study is the low number of selected patients?

Material and Methods:

The exception was that serum samples were diluted in 1:10 concentration in 
Red assay buffer. 

How did you decide to dilute 1:10 and no 1:100 for example?

Ethics

I strongly believe that you should mention Institutional Review Board Statement and that you received the  Informed Consent Statement

following the declaration of Elsinki

Author Response

Please find attached the file 

Reviewer 2 Report

Thank you for the opportunity to review this paper.

Overall, it is a pilot evaluation of a diagnostic test for detection of extracellular vesicles glycoconjugates associated to colorectal cancer (CRC). It is a very interesting issue because CRC is one of the most prevalent worldwide and increasing evidence suggests that extracellular vesicles could be used for early cancer detection, prognosis and to guide therapies.

The work is well structured and written, has up to date references and is of outstanding quality.

I have few suggestions:

In section 2.2 of Results, authors show the levels of significance between the different groups with the CD151-CD63 marker. Are there significant differences between the groups of patients' serum and those markers with the capacity to detect CRC in more advanced stages (CEA-ConA, CEA125-WGA, CA19.9-Ma696, CA19.9-ConA)? Although the heatmap of section 2.3 is a very visual way of data representation it is complex to assess their significance without statistical data.

Regarding the decrease of CD151-CD63 in late CRC. Don't the authors think that could represent a limitation for the use of the marker due to the possibility of false negatives in those patients between stages II-III?. Which of the advanced stage detection markers would the authors propose to use in combination with CD151-CD63? Have the authors tested this combination in patient’s serum?

The authors compare the discrimination capacity for classic markers such as CEA EIA, CA19.9 EIA or CA125 EIA and the new proposed markers. If you are trying to validate a new diagnostic test, it would be very interesting to see sensitivity, specificity, PPV-NPV  or accuracy analysis of these new markers compared to the classic ones.

At the end of the results, the authors argue that the combination of CD151-CD63 as well as other glycovariations can be used for follow-up of CRC patients as a marker of relapse. Have the authors validated this hypothesis with serum from patients with relapse? Given that combinations such as CD151-CD63 are only detectable in stages I-II of the disease, it would be interesting to verify the behavior of these new markers in the serum of patients with relapses.

Since the serum samples from healthy patients / benign disease come from different sources (donors) compared to patients with CRC (commercial serum). Has the processing of the samples been strictly the same in terms of handling, storage / freezing / thawing? Do the researchers know if the detection capacity of the markers varies depending on the storage/process of the samples?

Patients characteristics, table 1:

How do you select healthy patients? Do they have previous colonoscopy to be sure that they do not have any colonic disease or they are just asymptomatic?

Why do you choose Crohn´s disease? Do you think it should be interesting to have a group with benign colon polyps or even a subgroup of advanced adenomas?

In the CRC group, there is very high heterogeneity and the data is not clear.. I suppose that all of them are CRC, so why they are not 100% adenocarcinomas? If you have several with metastatic disease, why you don´t have any stage IV? T4N0M0 is not a stage IIIB, T3NX and T3N0 are not stage IIIA..

This main limitation for the study (the small number with high heterogeneity of the sample) must be shown clearer in the discussion.

Author Response

Please find an attached file 
